# 3D-Printed PLA Medical Devices: Physicochemical Changes and Biological Response after Sterilisation Treatments

**DOI:** 10.3390/polym14194117

**Published:** 2022-10-01

**Authors:** Sara Pérez-Davila, Laura González-Rodríguez, Raquel Lama, Miriam López-Álvarez, Ana Leite Oliveira, Julia Serra, Beatriz Novoa, Antonio Figueras, Pío González

**Affiliations:** 1CINTECX, Universidade de Vigo, Grupo de Novos Materiais, 36310 Vigo, Spain; 2Galicia Sur Health Research Institute (IIS Galicia Sur), SERGAS-UVIGO, 36213 Vigo, Spain; 3Institute of Marine Research (IIM), National Research Council (CSIC), 36208 Vigo, Spain; 4Universidade Católica Portuguesa, CBQF—Centro de Biotecnologia e Química Fina, Laboratório Associado, Escola Superior de Biotecnologia, Rua Diogo de Botelho, 1327, 4169-005 Porto, Portugal

**Keywords:** polylactic acid (PLA), 3D printing, medical devices, sterilisation, physicochemical changes, biological response, zebrafish model

## Abstract

Polylactic acid (PLA) has become one of the most commonly used polymers in medical devices given its biocompatible, biodegradable and bioabsorbable properties. In addition, due to PLA’s thermoplastic behaviour, these medical devices are now obtained using 3D printing technologies. Once obtained, the 3D-printed PLA devices undergo different sterilisation procedures, which are essential to prevent infections. This work was an in-depth study of the physicochemical changes caused by novel and conventional sterilisation techniques on 3D-printed PLA and their impact on the biological response in terms of toxicity. The 3D-printed PLA physicochemical (XPS, FTIR, DSC, XRD) and mechanical properties as well as the hydrophilic degree were evaluated after sterilisation using saturated steam (SS), low temperature steam with formaldehyde (LTSF), gamma irradiation (GR), hydrogen peroxide gas plasma (HPGP) and CO_2_ under critical conditions (SCCO). The biological response was tested in vitro (fibroblasts NCTC-929) and in vivo (embryos and larvae wild-type zebrafish *Danio rerio*). The results indicated that after GR sterilisation, PLA preserved the O:C ratio and the semi-crystalline structure. Significant changes in the polymer surface were found after HPGP, LTSF and SS sterilisations, with a decrease in the O:C ratio. Moreover, the FTIR, DSC and XRD analysis revealed PLA crystallisation after SS sterilisation, with a 52.9% increase in the crystallinity index. This structural change was also reflected in the mechanical properties and wettability. An increase in crystallinity was also observed after SCCO and LTSF sterilisations, although to a lesser extent. Despite these changes, the biological evaluation revealed that none of the techniques were shown to promote the release of toxic compounds or PLA modifications with toxicity effects. GR sterilisation was concluded as the least reactive technique with good perspectives in the biological response, not only at the level of toxicity but at all levels, since the 3D-printed PLA remained almost unaltered.

## 1. Introduction

Polylactic acid (PLA) is an aliphatic polyester produced as a racemic mixture of D and L lactide from nontoxic renewable sources, such as corn and sugarcane, with valuable properties for the biomedical field [1,2]. This polymer stands out for its behaviour in contact with biological media, as it gradually degrades into innocuous lactic acid or carbon dioxide and water and is metabolised intracellularly or excreted in urine and breath over time [1,3]. In addition to this immunologically inert response, PLA does not produce toxic or carcinogenic effects in local tissues, it is completely reabsorbed, and its production is relatively cost-efficient as compared to other traditional biodegradable polymers [4]. Given its biocompatible, biodegradable and bioabsorbable properties, PLA has become one of the most commonly used polymers in clinics with numerous applications including medical implants, porous scaffolds, sutures, cell carriers, drug delivery systems and a myriad of other fabrications [1,2,5].

Recently, with the emergence in the biomedical field of fused deposition modelling (FDM), one of the most common 3D printing techniques, interest in PLA has risen dramatically because of its favourable thermoplastic properties [6]. It can be heated to its melting point, cooled and reheated again without significant degradation. Together with FDM technology, this enables the rapid manufacture of customised structures and the fabrication of platforms for an extensive variety of applications both in research and in surgical practices, such as patient-specific implants, surgical guides (cranial and maxillofacial surgery) and surgical tools [5,6,7,8].

Once produced, the next critical step in the manufacturing process of these 3D-printed PLA-based medical devices is sterilisation, which is essential to prevent possible complications such as infections or rejections once in contact with the human body [9]. In this regard, despite the fact that PLA has been the focus of multiple pre-clinical and clinical trials, to what extent the different sterilisation techniques accurately affect its properties, and therefore its clinical functionality, continues to be the subject of debate, particularly now with the development of 3D printing [6]. It is clearly stated in the literature that conventional sterilisation techniques can cause physicochemical modifications in polymer-based medical devices that could limit their use in clinical applications [10]. Saturated steam is mostly not recommended for thermolabile and hydrolytically sensitive polymers—such as PLA—because of the high-temperature water vapour, which affects the structure of the sample prints. According to several authors [11,12], gamma radiation can induce chain scission or crosslinking reactions in polymers. Sterilisation by gas plasma with hydrogen peroxide is the recommended method for some authors, although it has an inferior penetrating capability compared to other methods [13]. On the other hand, low temperature steam formaldehyde sterilisation is widely used in European countries for the sterilisation of thermolabile medical equipment, but the residual levels of formaldehyde stipulated in the regulations must always be respected as it is known to be toxic and carcinogenic [14]. The same occurs with residues from ethylene oxide sterilisation as they may cause toxicity and induce a chemical reaction with the polymer matrix [10,11,13]. For these reasons, this methodology has been progressively prohibited by several hospitals in the EU and the USA [15]. The need to find new and effective sterilisation alternatives has brought the supercritical carbon dioxide methodology to the fore. This has emerged as a green and sustainable technology that requires moderate temperatures, avoiding physical and chemical damage in thermolabile and hydrolytically sensitive materials such as PLA and its derivatives [6,16]. It has already been proven to completely inactivate a wide variety of organisms [17] and even pores [18], however it is still a developing technology not yet optimised for every material, as is the case with PLA.

There is, therefore, a need to further study the extent to which sterilisation techniques affect PLA-based materials, preventing significant changes in their physicochemical, mechanical and biocompatibility properties that might give rise to adverse responses in the body or compromise bodily functions [12]. It must also be taken into account that the choice of sterilisation technique will be highly conditioned by the ease and cost-effectiveness of their implementation in the production routine. In fact, given the aforementioned evidence, it is expected that printed PLA, as a thermolabile and hydrolytically sensitive polymer, will be affected by conventional sterilization techniques such as saturated steam, which could in turn have effects on its biological performance.

Thus, the purpose of this paper was to study in detail the effect of the main sterilisation processes mentioned, both the current and emerging ones, on 3D-printed PLA samples in terms of their physicochemical (XPS, FTIR, DSC, XRD) and mechanical properties as well as their influence on the PLA wettability degree. Moreover, the biological response was also evaluated to determine whether the potential changes brought about by the different techniques significantly affect PLA behaviour in terms of toxicity when tested in vitro and in vivo. For the latter, the zebrafish model (*Danio rerio*) was used due to its advantages, as compared to conventional animal models, due to its close homology with the human genome including its immunogenic responses.

## 2. Materials and Methods

### 2.1. PLA Precursor Material

A 1.75 mm polylactic acid (PLA) filament 3D850, designed from the biodegradable resin formulation Ingeo™ PLA 3D850 by NatureWorks, was purchased from Smart Materials 3D, Jaén, Spain. In comparison to standard PLA, this improved biodegradable filament presents an elevated rate of crystallisation and very low thermal contraction, which makes printing more rapid and accurate without the risk of sample deformation. The main properties are summarised in Table 1.

### 2.2. Obtaining 3D-Printed PLA

Two sets of PLA samples were first designed using the SolidWorks 2016 software (Dassault Systemes SolidWorksCorp., Waltham, MA, USA), with one of discs measuring 1 mm in height and 5 mm in diameter, and the second consisting of well-shaped samples replicating the wells of 6-well microplates, measuring 34 mm in height and 16 mm in diameter. See Figure 1 for images from the software showing the two corresponding sets of samples. The digital data were then saved as STL files to subsequently generate corresponding sets of G-code for 3D printing using the Simplify3D software (Simplify3DSoftware, Cincinnati, OH, USA).

A dual extruder 3D printer (BCN3D+, 3D RepRapBCN, Barcelona, Spain) based on fused filament deposition modelling (FDM) technology was used to print both sets, and the printing temperature was maintained at between 190–230 °C. A concentric infill pattern was chosen in order to create a uniform structure and avoid gaps. The main 3D printing parameters are summarised in Table 2.

### 2.3. Sterilisation Techniques

PLA samples printed as discs were subjected to five different sterilisation treatments. Untreated 3D-printed material (PLA_C_) was also considered and analysed as control samples. The specific sterilisation treatments evaluated are described below:-Saturated steam (SS), carried out in an autoclave (Selecta Presoclave II 75, Cham, Switzerland) at Grupo de Novos Materiais (Vigo, Spain) operating at 121 °C and 2 bar for 20 min. The samples were sterilised in a surgical paper package.-Low temperature steam with 2% formaldehyde (LTSF), carried out in a Matachana steriliser (Matachana 130 LF, Matachana Group, Barcelona, Spain) at Povisa Hospital (Vigo, Spain). This steriliser complied with EN 14180:2014 and used a mixture of steam and 2% formaldehyde in thermodynamic equilibrium. Sterilisation was performed at 78 °C, with a standard duration of 153 min at full load and the samples were sterilised in a surgical paper package.-Gamma irradiation (GR), performed by Aragogamma S.L. (Barcelona, Spain) using a ^60^Co source irradiator at a dose level of between 25 and 35 kGy at room temperature, as per ISO 13485:2018.-Hydrogen peroxide gas plasma (HPGP), performed in a Sterrad NX steriliser (Advanced Sterilization Products, Irvine, CA, USA) at Hospital Universitario Lucus Augusti (Lugo, Spain). This was based on 59% aqueous hydrogen peroxide, which was concentrated to about 95% through removal of water from the peroxide solution before its evaporation and transfer to the chamber. PLA discs were conditioned in Tyvek packages compatible with the sterilisation process. The temperature during the sterilisation cycle was kept at between 45 °C and 55 °C in a standard cycle.-CO_2_ under critical conditions (SCCO) was carried out at the Biomaterials and Biomedical Technology lab (CBQF, Porto, Portugal). PLA discs were packed in sealed permeable plastic cartridges and placed in the reactor, to which hydrogen peroxide was added as an additive (300 ppm) to make sterilisation more effective. The optimised operating parameters for sterilisation were 40 °C temperature and 240 bar pressure with constant agitation of 600 rpm. Once pressurisation occurred, CO_2_ under critical conditions acted for 4 h.

The PLA discs/well-shaped samples subjected to the five sterilisation processes were named as follows: PLA_SS_, PLA_LTSF_, PLA_GR_, PLA_HPGP_ and PLA_SCCO_.

### 2.4. Physicochemical Characterisation

The elemental compositional analysis of PLA_C_, PLA_SS_, PLA_LTSF_, PLA_GR_, PLA_HPGP_ and PLA_SCCO_ discs was determined in detail by X-ray photoelectron spectroscopy (XPS) using a Thermo Scientific K-Alpha ESCA instrument (Waltham, MA, USA) equipped with an Al Kα monochromatised X-ray source radiation at 1486.6 eV (CACTI, UVigo, Vigo, Spain). Photoelectrons were collected from a take-off angle of 90° relative to the sample surface and the measurement was taken in a Constant Analyser Energy mode (CAE) with a 100 eV pass energy for survey spectra and 20 eV pass energy for high resolution spectra. Charge referencing was performed by setting the lower binding energy C1s photo peak at the 285.0 eV C1s hydrocarbon peak. The surface elemental composition was determined using the standard Scofield photoemission cross sections.

Chemical modifications in the polymeric chains of the PLA discs induced by the different sterilisation treatments were identified by Fourier Transform Infrared Spectroscopy (FTIR) using a Thermo Scientific Nicolet 6700 spectrometer (Waltham, MA, USA) with a DTGS KBr detector (CACTI, UVigo). Spectra were collected in the 400 to 4000 cm^−1^ wavelength range by averaging 34 scans and with a resolution of 4 cm^−1^.

To complement this information regarding the PLA chemical modifications, a thermal characterisation was also performed using a simultaneous TGA-DSC Setsys Evolution 1750 thermal analyser (Setaram, NJ, USA) in the CACTI, UVigo. The PLA discs were first subjected to a heating–cooling ramp (from 20 to 250 °C/min at 5 °C/min) with a constant supply of nitrogen. Secondly, a third ramp from 20 to 900 °C with a heating rate of 10 °C/min was applied with a constant supply of air. The information from DSC curves was used to determine the glass transition temperature (Tg) and melting point temperature (Tm), as well as the exothermal response relating to cold crystallisation (Tcc), which was obtained from the first heating cycle. The crystallinity index (Xc) was calculated according to Savaris et al. [19] based on the cold crystallisation enthalpy (ΔHcc), melting enthalpy (ΔHm) of the first heating calculated by peak fitting algorithms of the DSC curves and melting enthalpy of theoretically 100% crystalline PLA (ΔHm° = 93.7 J/g), in accordance with Equation (1):


(1)
$$\mathrm{Xc}\;\left(\%\right)=\frac{\mathrm{AHm}-\mathrm{AHcc}}{\mathrm{AHm}{}^{\circ}}\;\times \;100$$


The PLA crystalline structure was evaluated by X-ray diffraction (XRD) in an X’Pert Pro Panalytical diffractometer (Malvern Panalytical, Malvern, UK) with monochromatic Cu-Kα radiation (λ = 1.5406 Å) and with a 2θ range of 4–100° (40 kV, 30 mA, 0.013° step size) (CACTI, UVigo).

Contact angle measurements were performed in a Pocket goniometer (Fibro System AB, Stockholm, Sweden) at the Grupo Novos Materiais (UVigo) to evaluate the hydrophilic degree of the 3D-printed PLA discs. A sessile drop of ultrapure water was dispensed on each sample at room temperature and analysed using the linked software. The reported values corresponded to the average of ten measurements of each of the five replicates per treatment ± standard deviation. Images were taken immediately after the drop was deposited on the surface of the sample.

Finally, the mechanical properties were analysed using a nanoindenter XP (MTS Inc., Huntsville, AL, USA), in CACTI, UVigo. Hardness and Young’s Modulus values were measured using a 100 nm radius triangular pyramid indenter tip (Berkovich-type indenter) with the CSM (Continuous Stiffness Measurement) mode to perform dynamic measurements as a function of depth and XP head. A large number of indentations (30) were programmed, and the average of the valid results was calculated ± standard deviation.

Untreated 3D-printed PLA discs (PLA_C_) were also subjected to the above-mentioned physicochemical characterisation as control to be able to identify the changes caused by the various sterilisation techniques.

### 2.5. Biological Response In Vitro: Cytotoxicity Assay

To evaluate the cytotoxicity of the potential release of (1) small particles from the 3D-printed PLA after the different sterilisations (as a thermosensitive polymer) and of (2) potential traces of the toxic additives required in some of the sterilisation methods, a solvent extraction test was performed. It was carried out following the indications of UNE-EN-ISO 10993-5:2009 with the cell line NCTC clone 929 (ECACC 88102702) from mouse fibroblasts. The cells were incubated throughout the experiment at 37 °C in a humidified atmosphere with 5% of CO_2_ and with the cell growth medium DMEM (Lonza, Basilea, Switzerland), supplemented with 10% of foetal bovine serum (HyClone Laboratories LLC, Logan, UT, USA) and 1% of a combination of penicillin, streptomycin and amphotericin B (Lonza, Basilea, Switzerland).

To prepare the extracts, PLA_SS_, PLA_LTSF_, PLA_GR_, PLA_HPGP_ and PLA_SCCO_ discs were first placed in individual falcon tubes with the culture medium DMEM and kept at 37 °C for 24 h with 60 rpm agitation together with the controls. Then, different concentrations (100%, 50%, 30%, 10% and 0%) were prepared by diluting the initial extracts with fresh culture medium. A 6.4 g/L phenol solution was used as a positive control, while the negative control was the culture medium itself. The ratio between the material (PLA) and the volume of growth medium was 3 cm^2^ of material per ml of medium (ISO 10993-12).

A suspension of 1 × 10^5^ cells/m, in the same growth medium described above, were seeded in a 96-well microplate at a volume of 100 μL per well. After 72 h of incubation, a sub-confluent layer was formed, and the cell medium was replaced by the previously prepared extracts from 24 h before. Four replicates per concentration were incubated with the cells for 24 h. After that time, the cellular viability was quantified using the MTS Cell Proliferation Assay Kit (Abcam, Cambridge, UK). This colorimetric assay is based on the MTS tetrazolium compound, exclusively reduced by viable cells to generate a coloured formazan dye that is soluble in the culture medium. A volume of 10 μL of MTS reactive was added to each well. After 45 min of incubation, the reagent was renewed by fresh medium and the absorbance of the resulting solutions was read at a wavelength of 490 nm in a microplate spectrophotometer (Bio-Rad, Hercules, CA, USA). This test was repeated 3 times and the results were expressed as the percentage of viability compared to the negative control ± standard error.

### 2.6. Biological Response In Vivo: Acute Toxicity Test in Zebrafish

To evaluate in direct contact the in vivo toxicity caused by the physicochemical changes and/or toxic residues detected on the 3D-printed PLA after the sterilisation methods, a zebrafish model in direct contact was carried out and the corresponding 3D-printed well-shaped PLA samples were used. Embryos and larvae of wild-type zebrafish (*Danio rerio*, AB strain) were obtained from the experimental facilities at the Institute of Marine Research (IIM-CSIC, Vigo, Spain), where the animals were cultured and maintained following established protocols [20,21]. Adult zebrafish were maintained in a recirculating water system on a 12:12 h light–dark cycle and the water composition was maintained at pH 7.0 and 28–29 °C. The zebrafish were fed twice a day with commercial food (Nutrafin Max Tropical Fish Flakes) and once a day with live Artemia. Breeding crosses were conducted with a female/male ratio of 3:2. After the adult fish spawned, the embryos were obtained according to the protocols outlined in The Zebrafish Book [21] and maintained at 28 °C in an incubator (INE-500; Memmert, Schwabach, Germany) in zebrafish water during the whole experiment. All of the procedures in the experiment were reviewed and approved by the CSIC National Committee of Bioethics under approval number ES360570202001/17/FUN.01/INM06/BNG.

To evaluate the developmental toxicity of the zebrafish embryos, well-shaped PLA samples were placed on 6-well plates to prevent water loss. The wild type zebrafish fertilised embryos were collected and deposited in direct contact within these PLA samples (18 embryos per PLA well in triplicate for each condition) until the end of the experiment. First, the percentage of viable embryos was calculated at 24 hpf (hours post-fertilisation). Viable embryos were those that have live larvae inside and dead embryos became opaque, and this was also confirmed by the absence of a heartbeat under a Nikon SMZ800 microscope (Nikon, Tokyo, Japan). Once the non-viable embryos were removed from the well-shaped samples, the experiment continued to calculate the hatching rate at 24, 48 and 72 exposure hours by visual inspection and the survival larvae determined after 178 h (7 days). Mortality was assessed every 24 h by visual inspection discarding dead larvae, those that did not swim and those that did not have a heartbeat when viewed under a Nikon SMZ800 microscope (Nikon, Japan). Ten zebrafish larvae were randomly selected for each treatment at 3 dpf (days post-fertilisation), properly rinsed with zebrafish water and anesthetised with tricaine (0.05%) (Sigma-Aldrich, Merck Group, Darmstadt, Germany) to detect the main deformations under a Nikon AZ100 microscope (Nikon, Japan). Different measures, such as body length and yolk sac diameter, were also calculated with the aid of the ImageJ programme. In addition, the heart rate of embryos was counted at 3 days. Their heartbeat rates were measured from video recordings for 10 s at room temperature in set conditions. The data were analysed using a design macro in the Image J programme, which provided beats per minute. This allowed for the analysis of the heart rate of 3-dpf larvae. For all experiments, a blank control treatment without PLA was added in addition to the untreated PLA control material to validate the process. Each treatment was tested in triplicate and repeated twice, and all the results compared the mean values of the measurements after the treatments with respect to the control ± standard error.

### 2.7. Statistical Analysis

All biological data were analysed using GraphPad Prism 8 (GraphPad Software Inc., San Diego, CA, USA) and the results were represented graphically as the mean ± standard error of means (mean ± SEM). The nonparametric Mann–Whitney U test was used to determine the statistical differences between the control and the different sterilisation treatments. Statistical significance was determined to be * (*p* ≤ 0.05) at the 95% confidence level. In the zebrafish assay significant differences were obtained using the Mann–Whitney U test and displayed as *** (0.0001 < *p* < 0.001), ** (0.001 < *p* < 0.01) or * (0.01 < *p* < 0.05). Kaplan–Meier survival curves were analysed with a log–rank (Mantel–Cox) test.

## 3. Results and Discussion

### 3.1. Physicochemical Characterisation

The elemental composition of the surface of the PLA printed discs after sterilisation by the different techniques, PLA_SS_, PLA_LTSF_, PLA_GR_, PLA_HPGP_ and PLA_SCCO_, was first evaluated by XPS and presented as an atomic percentage (Table 3). The surface layer of the PLA printed disc without any sterilisation treatment (PLA_C_) was also subjected to the analysis and, as can be seen in Table 3, it was composed of a major contribution in C followed by O, in a ratio O:C of 0.50, with a minor contribution of Na. The PLA-tested discs where the elemental composition was more similar to the control were the PLA_SCCO_ and PLA_GR_, with an O:C ratio of 0.48 and 0.46, respectively. Conversely, the discs of PLA_LTSF_, PLA_HPGP_ and PLA_SS_ presented a relevant decrease in the O:C ratio to values in the range 0.35–0.27, which meant a lower oxygen content. Moreover, with these latter treatments, PLA_LTSF_, PLA_HPGP_ and PLA_SS_, a higher contribution of minor elements such as Si, Cl, S, I, Na and N was found compared to the two previous techniques. In relation to this, it is well known that in autoclave sterilisation, due to the high humidity conditions during the process, the transference of elements from the chamber to the sample surface is common. With respect to the small amounts of unexpected components, detected in the rest of the techniques, they can also be due to contaminations produced in the manufacture of the material and its subsequent handling (S, Cl, Na, N, C and O) or even the use of plastic gloves (Si, Na, S and Cl), with Si being the second most common contaminant in this technique (https://xpslibrary.com/contamination-2/) (25 July 2022). Finally, it is important to note that the differences found in the O:C ratios can be used as a guide to determine the degree of functionalisation of the polymer surface. Thus, the most significant change was detected in the autoclaved PLA discs (PLA_SS_) with a sudden increase in C at the expense of a decrease in the percentage of O, compared to the control (PLA_C_). On the other hand, the HPGP treatment led to a greater “oxidation” of the surface than the formaldehyde treatment (PLA_LTSF_). However, sterilisation by supercritical CO_2_ (PLA_SCCO_) and gamma radiation (PLA_GR_) caused only very slight changes at this level.

XPS high resolution spectra for C1s were taken to make an in-depth evaluation of the changes to the polymer by assigning energy binding peaks. Three binding energy assignments were obtained for all the tested PLA discs: C-H, C-C bonds at 285 eV, C-OH, C-O-C bonds around 287 eV and COOH, O=C-O bonds around 289 eV. However, differences were clearly detected. By way of example, Figure 2 shows the high-resolution spectra of the printed PLA control (PLA_C_) and PLA sterilised with saturated steam (PLA_SS_) where a significant change to the polymer surface after this latter treatment was clear. Thus, the C1s binding assignments obtained for PLA_C_ showed the three well-defined peaks with the assignments of C-H, C-C bonds being detected at higher intensity (285 eV) and the other two (at 287 and 289 eV) in slightly lower intensities attributed to C-OH, C-O-C and COOH, O=C-O, respectively. When the assignments obtained for the PLA_SS_ disc were observed, differences were clear with far less intense peaks at the binding energies at 287 eV and 289 eV and increased intensity (counts/s) for the assignment C-H, C-C at the binding energy 285 eV.

The C1s binding energy assignments with relative percentages for all the printed discs (shown in Table 4) indicated variations in the binding energy ranging from 287.10 eV for PLA_C_ to 287.20 eV for PLA_HPGP_ and PLA_SS_ and from 289.18 eV (PLA_C_) to 289.41 eV for PLA_LTSF_. The two sterilisation treatments that delivered more similar C1s binding assignments with respect to the control PLA_C,_ which presented C-H, C-C in 47.12 rel.%, C-OH, C-O-C in 27.35 rel.% and COOH, O=C-O in 25.54 rel.%, were PLA_SCCO_, firstly, followed by PLA_GR_. Meanwhile, PLA_SS_ and PLA_LTSF_ presented C-H, C-C in around 72 rel.%, C-OH, C-O-C in 17–18 rel.% and COOH, O=C-O in 9–10 rel.%. Both techniques, together with HPGP, caused a marked decrease in O and related bonds on the surface of the PLA.

The chemical changes in terms of vibrational modes were evaluated by FTIR spectroscopy to identify potential modifications in the characteristic absorption bands of PLA when subjected to the different sterilisation treatments. Figure 3 presents the FTIR spectra in the 400–2000 cm^−1^ wavenumber range, and the main absorption bands are identified. First, the absorption bands detected for the 3D-printed untreated disc, PLA_C,_ were evaluated and a correspondence with those of neat PLA fibres [22] was observed. This included a strong peak at 1747 cm^−1^ attributed to the stretching mode of carbonyl group (υ C=O) and a weaker one at 1266 cm^−1^ corresponding to the bending vibration of the same group (δ C=O). Second, major bands were recorded in the 1050–1200 cm^−1^ range, with three peaks at 1180, 1127 and 1080 cm^−1^ assigned to υ C-O vibrations. Bands in the 1300–1500 cm^−1^ range were also detected, and this was attributed to the deformational vibrations of C-H in the CH_3_ groups, with the asymmetric bending mode at 1452 cm^−1^ and the symmetric deformation at the doublet 1381–1360 cm^−1^ [23]. Other bands at 1043 and 868 cm^−1^, respectively, corresponding to δ O-H and υ C-C vibrations, were also identified.

The comparison between the FTIR spectra before (PLA_C_) and after the different sterilisation processes did not exhibit major differences, coinciding with the control PLA of 90–100% in the total wavenumber, 400–4000 cm^−1^. Indeed, the PLA_GR_ sample was the most similar to the control with a 99.7% fit, followed by PLA_SCCO_ at 97.3%, with the least similar to the control being the one sterilised by steam heat, PLA_SS_, 89.6%. Minor changes related to small chemical alterations were observed as slight shifts in some characteristic peaks. On the general PLA_C_ spectrum, the characteristic peaks of the υ C=O and δas CH_3_ bands were observed at 1747 and 1452 cm^−1^. Thus, for the PLA_SCCO_, PLA_LTSF_ and PLA_SS_ samples, an inversion of the intensities of the components of doublet 1380 and 1365 cm^−1^ and an increase in the signal at 1210 cm^−1^ were also observed and already reported [23]. These minor chemical alterations were clearly confirmed when the 960 to 830 cm^−1^ range was amplified and evaluated in depth (inset at Figure 3). According to the literature, this band is sensitive to the degree of crystallinity [24], where the crystalline α-phase corresponds to bands at 921 and 872 cm^−1^ and the amorphous phase with a band at 956 cm^−1^ [11,24]. The PLA_GR_ and PLA_HPGP_ spectra at the inset clearly presented only two peaks: at 872 cm^−1^ (crystalline α-phase) and 956 cm^−1^ (amorphous phase), exactly the same as with PLA_C_. In addition to these two peaks, in PLA_SCCO_, PLA_LTSF_ and PLA_SS,_ the crystalline α-phase peak at 921 cm^−1^ also appeared with higher intensity at the saturated steam samples, with PLA_SS,_ overcoming its amorphous band at 956 cm^−1^. The absence of the crystallisation peak at 921 cm^−1^ for the untreated PLA was in accordance with other authors [11,25]. Its presence for PLA_SCCO_, PLA_LTSF_ and PLA_SS_ suggests the crystallisation of the polymer to a greater or lesser extent for these latter three methods, as during crystallisation the 956 cm^−1^ band area decreased in a synchronised way with the appearance of the new band at 921 cm^−1^, characteristic of R crystals [26]. This tendency was confirmed by measuring the areas of both bands (956 cm^−1^/921 cm^−1^) with Magicplot software and a ratio of 921:956 was obtained for each PLA spectrum, with the obtained values for PLA_C_, PLA_GR_ and PLA_HPGP_ being 0.19, 0.16 and 0.21, respectively (no contribution of the crystalline peak at 921 cm^−1^). These values were slightly above 1 for PLA_LTSF_ and PLA_SCCO_ (1.13 and 1.14 respectively), which means an increased contribution of the crystalline peak to reach the same level as the amorphous one, and the highest value was for PLA_SS_ at 3.13 (clear higher contribution of the crystalline band in relation to the amorphous one). When revising the literature, Zhao and colleagues also describe the appearance of the band at 921 cm^−1^, characteristic of the crystalline α-phase at the same time as the decrease in intensity in the band at 956 cm^−1^ in PLA for disposable medical devices [10]. It is established that during sterilisation the presence of a high concentration of water, hydrogen peroxide or other small molecules can plasticise a thin layer on the surface of the polymer, facilitating crystallisation [25].

Next, the effect of the sterilisation on the thermal stability of the polymer was studied in detail using differential scanning calorimetry (DSC) in terms of how it influenced the degree of crystallinity. The first heat DSC thermograms for the different PLA samples are presented in Figure 4. The PLA_C_ thermogram showed the three distinct transitions typical of semi-crystalline thermoplastics: (1) heat flux at the glass transition temperature (Tg), (2) an exotherm associated with cold crystallisation (Tcc) and (3) a melting endotherm (Tm). A small exothermic band was observed immediately before melting occurred, associated with a pre-melt recrystallisation. Despite finding small differences in the values of the peaks of the three transitions, the thermograms obtained for the PLA_GR_, PLA_HPGP_ and PLA_LTSF_ samples presented the same semi-crystalline behaviour. This was not the case for the samples that were sterilised using autoclave and supercritical CO_2_ (PLA_SCCO_ and PLA_SS_), which presented the greatest changes. For the PLA_SCCO_ sample, the glass transition and cold crystallisation almost disappeared, which again suggests PLA crystallisation. In the same way, for PLA_SS_, these peaks were no longer observed, with only the melting peak (Tm) appearing, which is characteristic of crystalline polymers. These results supported the findings obtained from the FTIR analysis.

To confirm this, the corresponding crystallinity indices were calculated and are presented in percentage form (X_C_ %) in Table 5, together with peaks for key transitions. The first transition detected for PLA_C_ was a change in the specific heat of the material at the glass transition region at 63.2 °C immediately followed by another transition at 100.7 °C, which corresponded to cold crystallisation. This process was associated with the self-nucleation of the crystalline phases, where molecular chains that were preciously locked into position in the amorphous regions now had enough molecular mobility to reorganise into a more ordered and lower energy state crystalline phase. The final transition observed was a large endothermic peak at 175.5 °C associated with the heat of fusion for melting [25]. The index of crystallinity obtained was 18.5%. The peaks obtained in the key transitions were in accordance with other authors [1], with low glass transition temperature (Tg) values in the 60–65 °C range and a melting temperature (Tm) of 173–178 °C for PLA. When this untreated PLA_C_ was compared to the other samples, PLA_GR_ and PLA_HPGP_ were both observed to present a lower glass transition (Tg) in comparison to the control, PLA_C_. This could indicate some degradation of the amorphous regions of the polymer via hydrolysis or chain scission in these two types of sterilisation [27]. The degree of crystallinity obtained for these two samples was 15.1 and 17.4%, respectively, which was relatively close to the untreated PLA_C_ (18.5%). As expected after seeing the DSC thermograms in Figure 4, the PLA_SS_ sample only presented a melting endotherm transition at 176.8 °C and the highest degree of crystallinity with a value of 39.3%. During sterilisation with saturated steam, the crystallisation temperature of PLA was exceeded, crystallising the amorphous sample [10,19]. The same occurred for the PLA_SCCO_ and PLA_LTSF_ samples, although to a lesser extent than for autoclaving.

In accordance with our results, other authors [19] stated that gamma radiation can promote only slight changes in crystallinity by inducing ionisation reactions in the polymeric chains. At the same time, and again in line with our work, hydrogen peroxide gas plasma was proven to not chemically or morphologically affect PLA microfibers in electrospun scaffolds [23]. Moreover, the disappearance of a measurable glass transition (Tg), a cold crystallisation (Tcc) temperature and an increase in crystallinity in the DSC thermograms after PLA autoclave sterilisation has also previously been published [19]. This behaviour for PLA_SS_ was also observed for the PLA_SCCO_ samples (Figure 4), however previous works have established that supercritical CO_2_ sterilisation treatments did not induce changes in the calorimetric properties of macroscopic poly(L-lactic acid) porous scaffolds (on crystallinity and Tm) [18].

In order to supplement these results by identifying the PLA crystalline planes after the sterilisation treatments, the 3D-printed PLA samples were evaluated by XRD. Figure 5 shows the XRD diffraction patterns obtained for the most representative samples PLA_C_, PLA_GR_ (both at the inset) and PLA_SS_. The spectra obtained for PLA_C_ was of a semi-crystalline polymeric material. The same behaviour was observed after sterilisation of the sample by gamma radiation. However, the XRD profile of the saturated steam sample, PLA_SS_, exhibited two sharp diffraction peaks, a very intense one at 2θ = 16.6° and another at 19°, respectively attributed to (200)/(110) and (203) lattice planes. Weaker peaks were also detected at 12.4°, 14.7° and 22.3°, corresponding to the (004)/(103), (010) and (015) crystal planes, respectively. These peaks were very similar to those obtained by Zhao et al. [10] for PLA sterilised using saturated steam. In relation to the main crystalline form, it has been shown that PLA exhibits two different crystalline phases termed as α and α′. The latter is described as the disordered form of the stable α phase. According to Jalali and coworkers [28], for crystallisation processes at temperatures below 100 °C the main crystalline form present is the α′ phase, whereas above 120 °C it is the α phase, and within the 100–120 °C range a mixture of both crystalline phases is present. The same authors associated an exothermic peak at the DSC prior to the single melting temperature with the α′–α solid-state transition (100–120 °C), and a single melting of the α phase for samples crystallised above 120 °C. In our study, the crystallisation process referred to saturated steam sterilisation at 121 °C for 20 min and the DSC thermogram for PLA_SS_ (Figure 4) did not present any exothermic peak below the melting temperature. Taking all this into account, the α phase may have been the main crystalline form present.

The structural changes to the 3D-printed PLA samples caused by certain sterilisation methods may affect the mechanical properties of the biomaterial. The hardness and Young Modulus values were measured in a nanoindenter and are both presented in Figure 6. The results obtained for hardness of the control sample PLA_C_ presented a value of 0.27 ± 0.01 GPa, which was in agreement with other authors [29] with 0.23 ± 0.03 GPa. The sterilisation of PLA by gamma radiation did not affect the hardness of the polymer, which remained practically constant, which also occurred for PLA_SCCO_, PLA_HPGP_ and PLA_LTSF_. An intense increment was clearly detected when PLA was sterilised by autoclaving, PLA_SS_ (0.34 ± 0.03 GPa). This increase in hardness was in concordance with the increment in the crystallinity degree of the PLA [30]. In relation to the Young Modulus, the result obtained was 4.7 ± 0.1 GPa for PLA_C_, again in accordance with the literature [29] with 4.6 ± 0.4 GPa. A slight decrease was detected, and therefore in the PLA stiffness, after sterilisation by gamma radiation, supercritical CO_2_ and hydrogen peroxide gas plasma, and there was a steeper decrease with formaldehyde (PLA_LTSF_). Conversely, sterilisation by saturated steam promoted an increase in the value of the stiffness in PLA_SS_. It was, therefore, in this sample (PLA_SS_) where the physicochemical and structural changes seemed to be relevant enough to promote certain modifications at the mechanical behavioural level.

Finally, once the changes at the compositional, structural and mechanical level were evaluated, the wettability was also analysed, which is relevant to predicting how a biomaterial will behave within a biological environment. The contact angle measurements presented in Figure 7 confirmed the expected hydrophilic behaviour (<90°) of the untreated PLAC with a mean value of 72.1°. The PLAC contact angle obtained was in agreement with that measured by Savaris and colleagues (75.1°) [19]. When subjected to the sterilisation processes, slight modifications in the contact angle were evidenced, with the highest value being 79.3° for PLA_SS_, which was therefore the least hydrophilic sample in terms of the mean value. This latter sterilisation caused a decrease in the amount of oxygen at the printed PLA surface (Table 3). Hydrophilicity is generally related to oxygen concentration and the presence of polar species such as carbonyl, carboxyl and hydroxyl groups [31]. A slight decrease in wettability was then expected for PLA_SS_ sterilisation. Significant variability was detected for the other samples, PLA_SCCO_, PLA_HPGP_, PLA_GR_ and PLA_LTSF_, with the lowest angle measured being in the latter (61.9°). According to the amount of oxygen in the surface of the PLA_LTSF,_ the expected decrease in wettability did not occur, and the sterilisation even favoured it. More exhaustive studies on this specific technique could be performed, as the wettability is influenced by several factors, not only oxygen. When performing our experiment, the contact angle was measured at the smooth part of the 3D-printed PLA disc (the one in contact with the hot bed), as the side with the semi-circular pattern gave very unstable values. 

In summary, the analyses carried out showed that the physicochemical properties of PLA can be affected to a greater or lesser extent depending on the sterilisation technique used. Compositionally, gamma radiation did not contribute with external minor elements, providing a similar O:C ratio with respect to the control sample PLA_C_. Conversely, autoclave sterilisation presented a relevant decrease in the O:C ratio and the highest contribution of minor elements (Si, S, N, Na and I). Moreover, due to the high temperatures and pressure in the PLA_SS_ process, some degree of crystallisation took place, demonstrated by the results of the analyses carried out using FTIR, XRD and DSC techniques, as indicated by other authors [13,32]. This change in crystallinity was also observed in the PLAscco and PLA_LTSF_ samples, although to a lesser extent. Other techniques, such as gamma radiation, maintained the original semi-crystalline structure of the unsterilised material (PLA_C_). In terms of hardness and Young Modulus, PLA_GR_, PLA_SCCO_, PLA_HPGP_ and PLA_LTSF_ presented minor changes when compared to the PLA_C_. However, saturated steam sterilisation promoted an increment in the hardness properties of the PLA_SS_, in concordance with the increment in its crystallinity degree. Changes to the wettability revealed the expected hydrophilic behaviour for all the tested samples with, again, the highest mean value of the contact angle being that for PLA_SS_, which implies the lowest hydrophilic degree. Saturated steam sterilisation applied to 3D-printed PLA samples was then clearly proven to promote changes to the physicochemical properties of the polymer by increasing its crystallisation and, to a lesser extent, the super critical CO_2_ technique, which is still experimental and surely needs the parameters to be optimised for these types of polymers. On the contrary, gamma radiation was clearly the least reactive with the biomaterial causing barely any alterations to it. To transfer these results into clinical practice, it is important to take into account that some methods are easier or more viable to implement than others. Furthermore, it is of great relevance to assess the extent to which the changes promoted in the biomaterial by the different sterilisation methods significantly affect its biological response. This aspect was tested first in vitro and later in vivo using the zebrafish model.

### 3.2. Biological Response: Cytotoxicity In Vitro and Acute Toxicity in Zebrafish Model

The in vitro evaluation of the potential cytotoxicity caused by the physicochemical changes detected in the 3D-printed PLA discs after the different sterilisation methods is presented in Figure 8. Given the absence of sterilisation in the control PLA, this sample could not be tested, and the sample extracts tested were from PLA_GR_, PLA_SCCO_, PLA_LTSF_, PLA_HPGP_ and PLA_SS_ in a wide concentration percentage from 0 to 100%. All the experiments were compared with a negative control, which was the same culture medium without a sample extract, and a positive control, represented by phenol. A total of 100% of the extract (black) for all the PLA samples tested presented a lower mean value of cell viability than the negative control, however these differences were not found to be statistically significant for any of them, even for 100% of PLA_SS._ The only PLA sample where the mean cell viability values for the different extracts were higher than the negative control were 50%, 30% and 10% of PLA_GR_ extract. However, when considering the standard deviation, the values were again not found to be statistically significant. Under all the test conditions, all values were above 80% of cell viability and therefore non-cytotoxic, being above the cytotoxicity limit of 70% established by the standard (UNE-EN-ISO 10993-5:2009). The cytotoxicity found at the 100%, 50% and 30% extracts of the positive control (phenol) validated the experiment. An absence of cytotoxicity in the PLA_GR_, PLA_SCCO_, PLA_LTSF_, PLA_HPGP_ and PLA_SS_ extracts towards fibroblasts NCTC clone 929 was demonstrated.

Once the absence of cytotoxicity in vitro had been proven, the 3D-printed PLA well-shaped samples of the sterilisation processes that triggered the most significant physicochemical changes (PLA_SS_) and the least significant (PLA_GR_), together with untreated PLA (PLA_C_), were evaluated in direct contact with the zebrafish model for the in vivo evaluation of potential acute toxicity. Figure 9 shows an optical image of the 3D-printed PLA well-shaped samples sterilised by the five techniques and the control, where a clear change in the colour of the PLA was observed. The material turned from transparent to white in the PLA_SS_ samples, and to a lesser extent with PLA_SCCO_. According to the literature and in agreement with our results, Weir et al. [32] clearly stated that when PLA crystallises it becomes opaque. Moreover, a certain deformation was clearly observed for PLA_SS,_ with such a deformation after the autoclaving process in 3D-printed parts having previously been reported by several authors [13].

Figure 10 displays the development and growth of zebrafish embryos on the 3D-printed PLA samples PLA_SS_, PLA_GR_ and PLA_C_. The figures show viable embryos (%) after 24 h (A); hatching rate (%) at 24, 48 and 72 hpf (B); survival rate of larvae after 7 dpf (C) and larvae at 3 dpf on the PLA_C_ and PLA_SS_ well samples (D). As already discussed, for the acute toxicity experiments in the zebrafish model, a blank control treatment without PLA was added in addition to the untreated PLA control material to validate them. Given that similar results with non-significant differences were observed for the two controls, the results presented in Figure 10, and at the evaluation of the main morphological structures of 72 hpf larvae at Figure 11, are represented only against the untreated PLA_C_. These similar results in both controls proved, in the first place, that PLA_C_ itself had no effect on the hatching or survival rates. Secondly, viable embryos were counted after 24 hpf (Figure 10A) and the embryos exposed to both PLA_GR_ and PLA_SS_ showed a percentage of viability similar to the control. The hatching rates of embryos exposed to the materials at 24, 48 and 72 h were also compared (Figure 10B). Following their normal development, the embryos began to hatch on the second day (48 h), with no significant differences between the two treatments compared to the control. At 72 h, all viable embryos had fully hatched. The rate was not delayed for the embryos exposed to PLA samples sterilised by gamma radiation or saturated steam. In the case of the larvae, the mortality rate was estimated in a continuing observation period every 24 h to acquire the survival percentage (Figure 10C). For the two sterilisation processes tested, the mortality rate did not present significant differences from the control.

An assessment of the toxic effects on embryonic developmental stages was also carried out, and the main morphological structures of the 72 hpf larvae zebrafish models were inspected under a microscope. Figure 11 represents the images of the normal body shape morphology of 72 hpf larvae after exposure to PLA_C_ (A), PLA_GR_ (B) and PLA_SS_ (C) together with body length (D), yolk sac diameter (E) and heart rate of 72 hpf larvae (F). Images B and C show that the body shape of the larvae at 3 dpf was completely normal after direct exposure to the PLA_GR_ and PLA_SS_ samples, respectively, and compared to the control group. None of the most common morphological defects, which would imply some degree of cytotoxicity, were observed, such as pericardial oedema, yolk sac oedema, ocular oedema, spine curvature, head malformation, shortened body stature, lowered yolk consumption and underdeveloped brain and eye [33]. To verify these visual inspection results, measurements of the body length (Figure 11 bottom left) of the larvae were taken, presenting a mean of 3.8 mm in the control sample PLA_C_. This value was very close to that described by Nüsslein et al. of 3.5 mm at 3 dpf, validating our PLA_C_ control [20]. For the PLA_GR_ and PLA_SS_ samples, the values were very close to the control with no significant differences, indicating that there was no delay in larval growth. The same was repeated for the yolk sac diameter, with no significant differences compared to the control. The yolk sac was completely normal in all three conditions, since no oedema or increase in size was observed that would indicate a lowered yolk consumption. Finally, the heart rate (Figure 11 bottom right) can be used as an indicator of the state of cardiac function of the embryo, which is an important parameter for studying toxicity. Due to the transparency of the embryos, the heart development, heart rate and deformations can be seen at a simple resolution [33]. The heartbeat ratios of the embryos exposed to PLA sterilised by gamma radiation and autoclaving were counted and presented no significant differences between them or as compared to the control (130 beats/minute). Moreover, the results were in agreement with the values of other studies at 133.08 beats/minute [34]. These results revealed that direct exposure to the PLA sterilised using these techniques did not bring about any changes in heart rate such as brachycardia or tachycardia. There was no toxic influence on the zebrafish hearts.

In summary, the in vivo evaluation proved that direct contact exposure to both the 3D-printed PLA_GR_ and the PLA_SS_ well-shaped samples did not induce embryonic developmental abnormalities and was not toxic to developing zebrafish embryos or early larvae stages. Other authors have used the zebrafish model to study the toxicity of different printing methods on zebrafish embryos, but with other materials (ABS) [35] and only cleaning the parts according to the manufacturing specifications [36]. As indicated by Zhu and colleagues, a limited number of research works have been completed on the toxicity of 3D materials to cells, tissues and model organisms. For this reason, they evaluate the toxicity of different polymers used in the 3D printing processes, with rapid methods to test for toxicity, including zebrafish assays. Their results for PLA_C_, sample extracts and direct contact of PLA with embryos were also non-toxic, however they did not indicate any sterilisation method [37,38].

## 4. Conclusions

The present work demonstrated that 3D-printed PLA remained practically unaltered when sterilised using gamma irradiation in terms of physicochemical and elemental composition, O:C ratio and distribution of binding assignments, maintaining also the original semi-crystalline structure of the unsterilised material. Conversely, saturated steam sterilisation was proven to generate significant physicochemical changes to the 3D-printed PLA, beginning with a 54% reduction in the O:C ratio caused by an intense decrease in O and related bonds. Moreover, an increase in the crystallinity degree of the 3D-printed PLA was demonstrated in the form of α-phase R crystals, confirmed by thermogravimetry, with an FTIR band ratio of 921:955 cm^−1^ of 3.13, which implied a clear contribution of the band sensitive to the degree of crystallinity (921 cm^−1^), a 52.9% increase in the index of crystallinity and the diffraction pattern. These changes, promoted by saturated steam sterilisation, slightly affected the mechanical properties and hydrophilic degree of the 3D-printed PLA. The three remaining sterilisation techniques evaluated (hydrogen peroxide gas plasma, low temperature steam with formaldehyde and CO_2_ under supercritical conditions) were situated at the intermediate point in terms of physicochemical changes. In fact, a decrease in the O:C ratio was also found after sterilisation using hydrogen peroxide gas plasma and low temperature steam together with a change in crystallinity after sterilisation using CO_2_ under critical conditions and low temperature steam with formaldehyde. Finally, the present work also demonstrated that despite all these changes the biological response of the 3D-printed PLA was not affected in terms of toxicity for any of the tested sterilisations, either in vitro or in the zebrafish animal model. However, GR sterilisation was clearly concluded as the least reactive technique with good perspectives in terms of the biological response, not only at the level of toxicity but also in terms of biofunctionality.

## Figures and Tables

**Figure 1 polymers-14-04117-f001:**
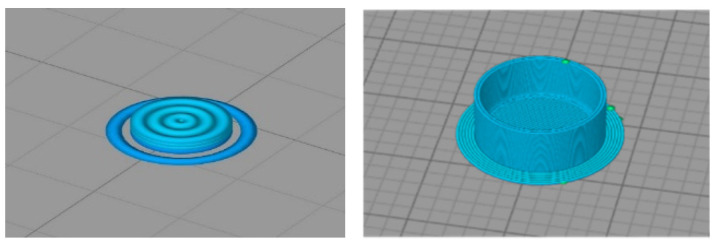
Software images showing the design of the PLA discs and the well-shaped samples.

**Figure 2 polymers-14-04117-f002:**
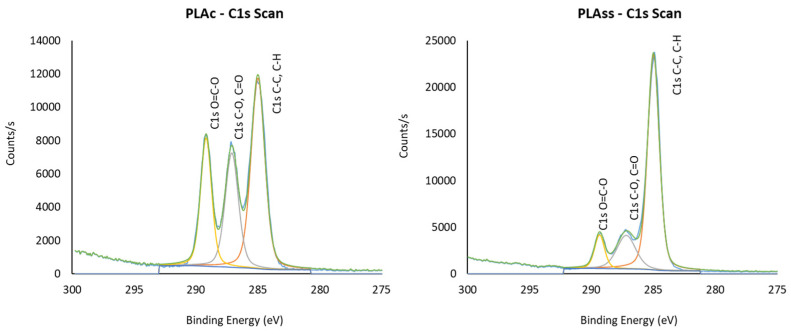
XPS high resolution spectra for C1s of printed PLA control (PLAC) and PLA sterilised with saturated steam (PLA_SS_).

**Figure 3 polymers-14-04117-f003:**
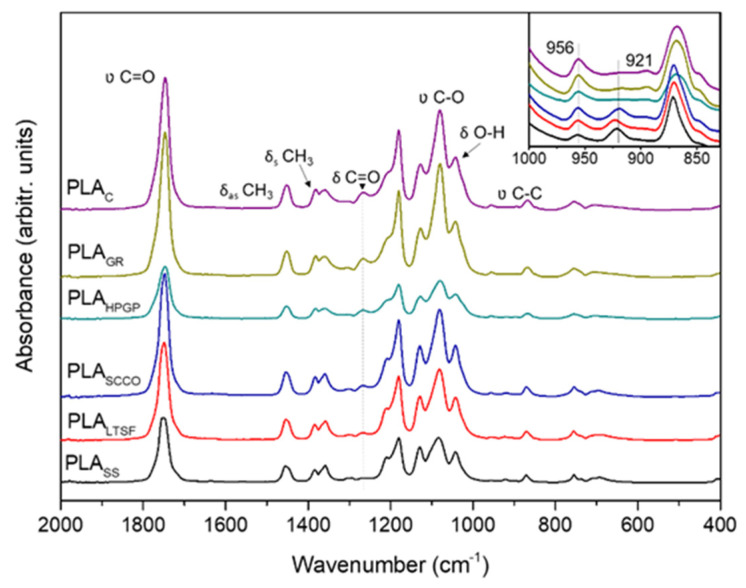
FTIR spectra of the PLA discs subjected to the different sterilisation treatments with absorption bands identified: PLA_GR_, PLA_SCCO_, PLA_LTSF_, PLA_SS_ and PLA_HPGP_ together with the untreated PLA disc as control, PLA_C_. Amplification of the 400–2000 cm^−1^ wavenumber range in the inset with the two main peaks identified.

**Figure 4 polymers-14-04117-f004:**
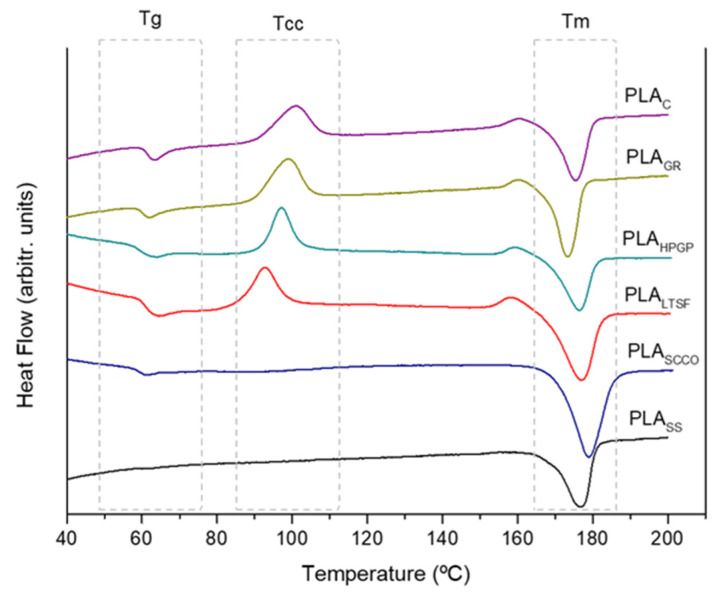
DSC thermograms for PLA discs subjected to the different sterilisation treatments: PLA_GR_, PLA_SCCO_, PLA_LTSF_, PLA_SS_ and PLA_HPGP_ together with the untreated PLA disc as control, PLA_C_.

**Figure 5 polymers-14-04117-f005:**
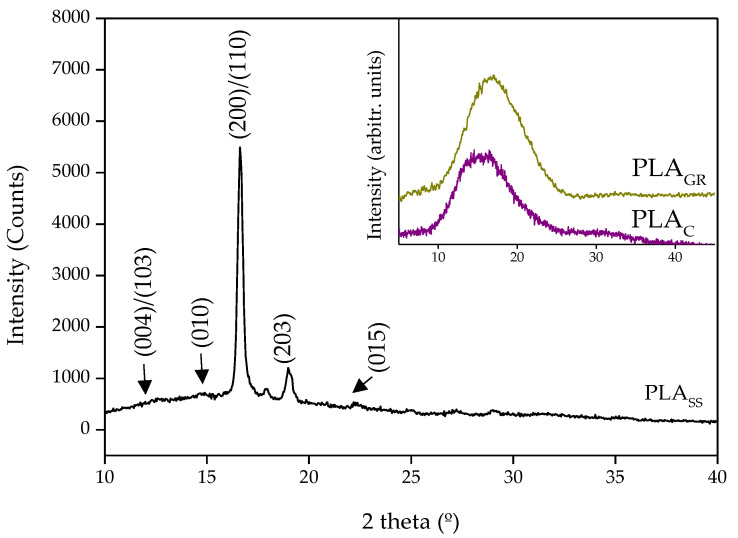
XRD diffraction pattern of untreated PLA_SS_ with the identification of the characteristic peaks and PLA_C_, and PLA_GR_ diffraction pattern at the inset.

**Figure 6 polymers-14-04117-f006:**
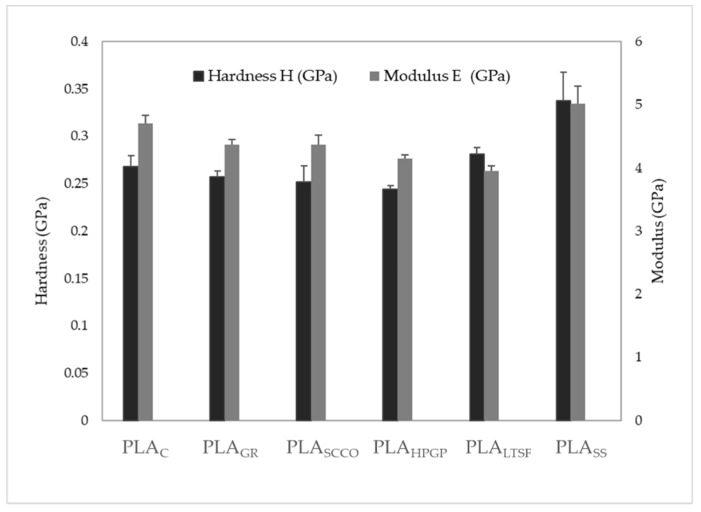
Modulus and hardness of PLA discs subjected to the different sterilisation treatments: PLA_GR_, PLA_SCCO_, PLA_LTSF_, PLA_SS_ and PLA_HPGP_ together with the untreated PLA disc as control, PLA_C_. Results are presented as mean ± standard deviation.

**Figure 7 polymers-14-04117-f007:**
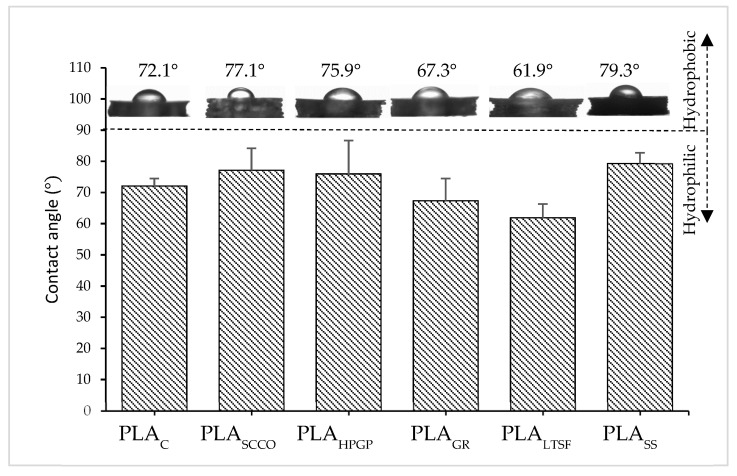
Water contact angle for PLA discs subjected to the different sterilisation treatments: PLA_GR_, PLA_SCCO_, PLA_LTSF_, PLA_SS_ and PLA_HPGP_ together with the untreated PLA disc as control, PLA_C_. Results are represented as mean ± standard deviation.

**Figure 8 polymers-14-04117-f008:**
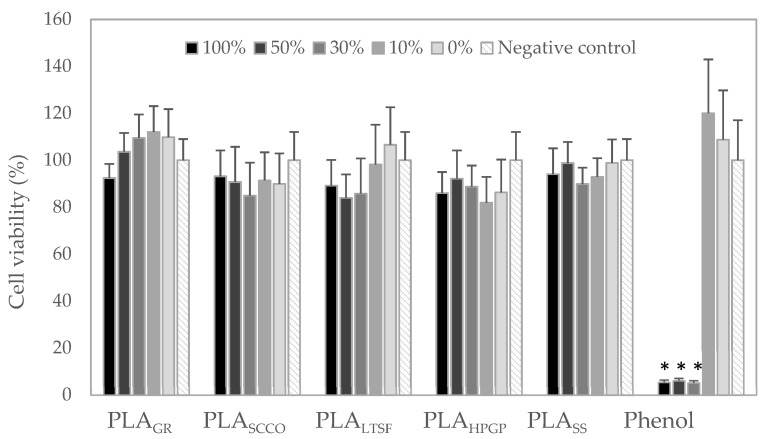
Cell viability detected in mouse fibroblast cells (NCTC clone 929) in the presence of different extracts of PLA_GR_, PLA_SCCO_, PLA_LTSF_, PLA_HPGP_ and PLA_SS_ discs, compared to the positive control (phenol) and negative control (DMEM). Results are expressed as percentage compared to the negative control ± standard error. Statistical significance was determined to * (*p* ≤ 0.05).

**Figure 9 polymers-14-04117-f009:**
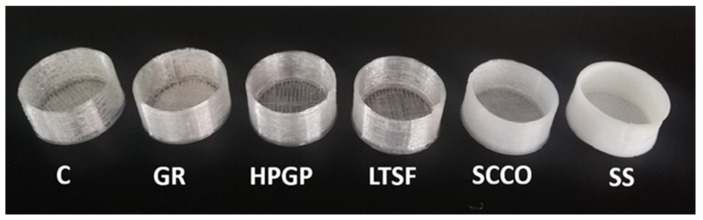
Image of 3D-printed PLA well samples before (untreated control C) and after five sterilisation processes: (GR) gamma radiation, (HPGP) hydrogen peroxide gas plasma, (LTSF) low temperature steam formaldehyde, (SCCO) supercritical CO_2_ and (SS) saturated steam.

**Figure 10 polymers-14-04117-f010:**
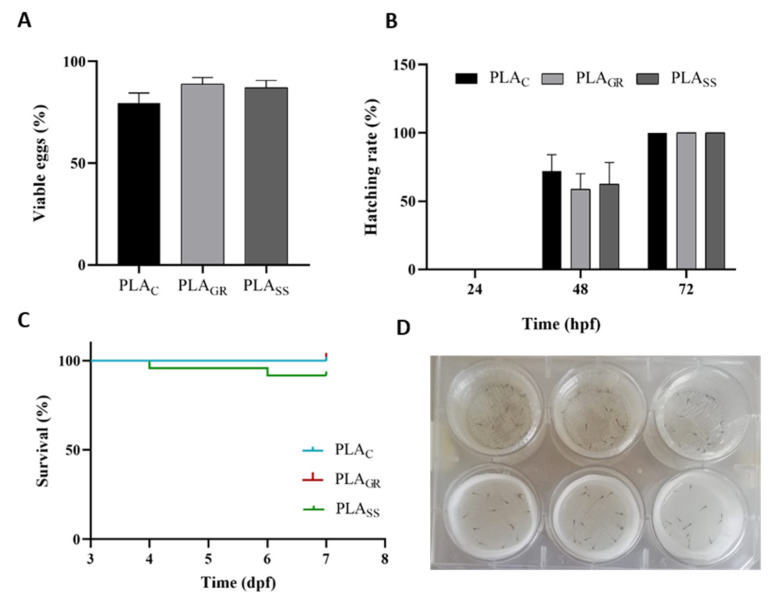
Viable embryos (%) after 24 h after exposure to PLA_GR_, PLA_SS_ and PLA_C_ (**A**); hatching rate (%) at 24, 48 and 72 hpf (**B**); survival rate of larvae after 7 dpf (days post-fecundation) (**C**); larvae at 3 dpf on the PLA_C_ (above) and PLA_SS_ (below) well samples (**D**). Results are represented as the mean ± standard error.

**Figure 11 polymers-14-04117-f011:**
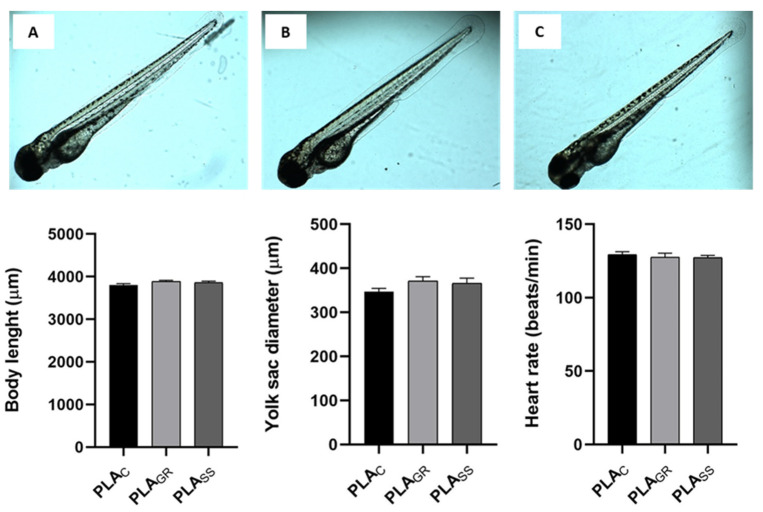
(**A**–**C**): Images of normal body shape morphology of 72 hpf larvae after exposure to PLA_C_, PLA_GR_ and PLA_SS_. Body length, yolk sac diameter and heart rate of 72 hpf larvae after exposure to PLA_C_, PLA_GR_ and PLA_SS_. Results are represented as the mean ± standard error.

**Table 1 polymers-14-04117-t001:** Technical data of PLA 3D850 filament taken from the Smart Materials 3D website: www.smartmaterials3d.com/pla-3d850 (2 August 2022).

Polylactic Acid 3D850	Value
Material density	1.24 g/cm^3^
Tensile yield strength	65.5 MPa
Flexural strength	126 MPa
Flexural modulus	4357 MPa
Heat distortion temperature	144 °C
Extrusion temperature	190–230 °C

**Table 2 polymers-14-04117-t002:** Main 3D printer parameters used.

Dual Extruder BCN3D+ Printer	Parameter Value
Nozzle	0.4 mm
Nozzle temperature	190–230 °C
Bed temperature	45 °C
Infill density	100%
Infill pattern	Concentric
Speed	60 mm/s
Layer height	0.2 mm

**Table 3 polymers-14-04117-t003:** Elemental composition of untreated PLA and after sterilisation treatments obtained from XPS analysis (at. %).

Samples	C	O	Si	S	N	Na	I	Cl	O/C
PLA_C_	66.24	33.42	-	-	-	0.34	-	-	0.50
PLA_GR_	68.65	31.35	-	-	-	-	-	-	0.46
PLA_SCCO_	66.52	31.82	-	-	1.24	0.42	-	-	0.48
PLA_LTSF_	73.78	21.20	3.34	-	1.07	0.61	-	-	0.29
PLA_HPGP_	70.92	24.73	1.85	-	1.11	0.93	-	0.47	0.35
PLA_SS_	74.94	20.00	1.23	0.18	2.53	0.92	0.19	-	0.27

**Table 4 polymers-14-04117-t004:** Carbon binding energy (BE) assignments with relative percentage (rel.%) from high resolution XPS spectra of untreated PLA and PLA after sterilisation treatments.

	C-H, C-C		C-OH, C-O-C		COOH, O=C-O	
	BE	Rel.%	BE	Rel.%	BE	Rel.%
PLA_C_	285	47.12	287.10	27.35	289.18	25.54
PLA_GR_	285	54.45	287.12	23.95	289.20	21.61
PLA_SCCO_	285	49.62	287.13	27.55	289.20	22.83
PLA_LTSF_	285	72.64	287.18	17.32	289.41	10.04
PLA_HPGP_	285	67.36	287.20	18.75	289.35	13.89
PLA_SS_	285	72.06	287.20	18.29	289.36	9.66

**Table 5 polymers-14-04117-t005:** Glass transition temperatures (Tg), cold crystallisation (Tcc), crystalline melting (Tm) and crystallinity index of PLA subjected to the different sterilisation treatments: PLA_GR_, PLA_SCCO_, PLA_LTSF_, PLA_SS_ and PLA_HPGP_ together with the untreated control, PLA_C_.

	Tg (°C)	Tcc (°C)	Tm (°C)	Xc (%)
PLA_C_	63.2	100.7	175.5	18.5
PLA_GR_	61.9	98.6	173.3	15.1
PLA_HPGP_	59.1	97.1	176.4	17.4
PLA_LTSF_	66.6	92.9	176.9	29.0
PLA_SCCO_	61.1	99.4	178.9	32.6
PLA_SS_	-	-	176.8	39.3

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
