# Peer review of "3D-Printed PLA Medical Devices: Physicochemical Changes and Biological Response after Sterilisation Treatments"

_polymers, 2022, doi:10.3390/polym14194117_

Round 1
Reviewer 1 Report
This is really an interesting paper, how the sterlization affects the functions of PLA in practically. However, there are few issues the authors need to address before further process.
Specific comments
Recently its use has risen: Reframe
as well as their influence on the PLA wettability: Define ´their´?
Poor description of results in abstract. Should briefly explain the details of results bit more in abstract
Gamma irradiation was clearly proven as: Briefly explain how did you come to this conclusion in abstract?
while consistent differences: Explain
demonstrated for several of the sterilisation methods: Reframe, not clear
This was particularly: this means?
Provide clear conclusion in abstract with optimum sterilization technique and parameters. Also clearly explain the changes in 3D PLA matrix during different sterilization technique.
-Low temperature steam with 2% formaldehyde: Explain bit more here. How did you use formaldehyde for sterilization at low temp? fill the samples with formaldehyde or any other way? package used?
-Hydrogen peroxide gas plasma: Same comment as above. Hydrogen peroxide added with samples in package as additive?
nano-mechanical properties were analysed using a nanoindenter NanoXP (MTS NanoSystem, CACTI UVigo). Hardness and Young's Modulus values: What did you mean nano-mechanical properties? Are you dealing with any nano material here or Hardness and Young's Modulus values are nano-mechanical properties?
PLASS, PLALTSF, PLAGR, PLAHPGP and PLASCCO discs were first placed: Why you did not use ´Untreated 3D printed PLA discs (PLAC)´ in cell culture experiment as control?
kept at 37 °C for 24 hours to obtain the corresponding extracts: Why did you want to get this extract? what you are expecting to get in this extract? Did you think that the PLA matrix is soluble in cell growth medium DMEM?
obtain the corresponding extracts: Quantify the target active substances in this extract that you are expecting to alter the cell growth.
falcon tubes with the cell growth medium DMEM (Lonza): First explain about the cells name, source, culture condition and no of passage used.
a suspension of... the same growth medium were: Grammar
cell medium was replaced by the previously prepared extracts: This is my concern, why dont you culture the cells directly on your samples instead of using condition medium? What was the actual active substance in this extract that alter the cell growth? Explain how you stored the extract?
Cell Proliferation Assay Kit (Abcam).: City and country
After 45 minutes of incubation, the absorbance: Why dont you remove the unbound excess reagent? it gives false positive result
Four replicates per concentration... test was performed in triplicate: Contradictory
Four replicates per concentration: Not clear. List the concentration here.
To evaluate in vivo the potential acute toxicity: Not clear
well-shaped PLA samples were placed in direct contact with the embryos on 6-well plates: What you mean direct contact? I think the authors placed the PLA samples in 6 well plate and added the embryos inside the well-shaped PLA samples, right? If so, explain this clearly.
deposited on the samples: PLA sample right?
18 embryos per well in triplicate: here PLA well or 6-well? Not clear how many PLA-wells were palced in each well of 6-well plate, so need more information here.
the percentage of viable embryos was calculated: Explain how? describe the technique
. Once the non-viable embryos were removed: How did you confirm the viability of embryos?
, the experiment continued to calculate the hatching rate: Here need to describe bit more about the experiment, how did you culture the embryos in 6-well plate? what was the medium and supplement used? Culture temperture? Incubator used?
Mortality was assessed every 24 hours.: How by visual inspection?
Biological response in vivo: Acute toxicity test in zebrafish: Overall, from the described method, the authors hatched zebrafish embryos on well-shaped PLA and calculated the hatching rate at 24, 48 and 72 exposure hours. As per the method, the PLA well was used like a vessel to grow the embryos, how did this really reflect the toxicity? Did you think the PLA liberate any substance to embryos through the culture medium? What was the biological relevance?
Table 3: Provide SD value and p value. Also explain the proper reason for observing new elements such as Si, S, N, I, and Cl in specific samples in text?
and/or being handles: explain
, Figure 1 shows: Figure 2 not 1
Figure 2. XPS high resolution spectra for C1s of printed PLA control (PLAC) and PLA sterilised with 310 saturated steam (PLASS): Why specifically PLASS? not other samples? Provide the date for other samples as well.
Figure 3: Label y axis value, explain the insert details in legend.
Figure 4: Label y axis value
Figure 5. XRD diffraction patterns of untreated PLA (PLAC), and PLAGR (both at the inset) and after saturated steam sterilisation, PLASS.: Not clear caption. Explain what was the XRD spectra with arrows and insert?
after saturated steam sterilisation, PLASS: So the PLAGR at the insert in before sterilization? Why listed only 3 samples here? provide the data for all six samples PLAC, PLASS, PLALTSF, PLAGR, PLAHPGP and PLASCCO. Do this in all results you presented.
Figure 6.: Why did you label y axis as hardness (GPa), since there was Modulus E in the same image. Provide p value with * in image
Figure 7. Provide the image with degree angle in each image and statistics * on bars
in a wide concentration percentage from 0 to 100%: mention this in M&M section.
limit of 70% established by the standard.: Reference
Figure 10: I am not sure this is the right technique to actually determine the toxicity of PLA material. Recently, folks are trying to use PLA based 3D matrix for human therapy, and the used technique by authors did not reflect the scenario in practically.
Figure 10. Larvae at 3 dpf on the PLAC and PLASS well samples (D): why no PLAGR here? Label in image which is which one. there were six pictures for two samples, why?
Author Response
The comments have been uploaded as a pdf file. Please see the attachment. Thank you.

Reviewer 2 Report
The present study focused on 3D printed PLA medical devices: physicochemical changes and biological response after sterilisation treatment. In my opinion, this manuscript is well written, the issue is interesting and the experiment is complete. The following points need to be revised, and after that, it is suggested to publish in Polymers.
1. Please confirm whether the article is in American English or British English. (e.g., sterilisation or sterilization).
2. The Introduction covers the complete message, and background and has a clear purpose and motivation. However, the hypothesis of this experiment is not described, please add the experimental hypothesis.
3. Materials and Methods, the author clearly describe all practical steps but please added the manufacturer and the country of the materials and equipment used. On the other hand, the statistical analysis must first confirm normal distribution and homogeneity and then can confirm whether the one-way ANOVA or Student’s t-test can be used.
4. Regarding the Biological response, what is the reason for using mouse fibroblast cell line 209 NCTC clone 929?
5. XPS results, why the PLAGR without any contribution of Na? the relative percentage (%) of C-OH, C-O-C, COOH, O=C-O for PLALTSF, PLAHPGP, and PLASS are lower, what is the reason?
6. The author state that due to the semi-circular pattern the values of contact angle are unstable. What I am curious about here is what is the reason for the difference in the hydrophilicity and hydrophobicity of the material surface after different sterilization treatments. Because the contact angles of some samples are increasing and some are decreasing; yet, their appearance seems to be the same.
Author Response

(The authors gave the same response as above.)

Round 2
Reviewer 1 Report
The revised version is satisfactory.
Reviewer 2 Report
The appended changes are satisfactory for acceptance.